# The Current Status of T Cell Receptor (TCR) Repertoire Analysis in Colorectal Cancer

**DOI:** 10.3390/ijms26062698

**Published:** 2025-03-17

**Authors:** Hiroyuki Takahashi, Katsuzo Hanaoka, Hideo Wada, Daibo Kojima, Masato Watanabe

**Affiliations:** Department of Surgery, Fukuoka University Chikushi Hospital, 1-1-1 Zokumyoin, Chikushino 818-8502, Fukuoka, Japan; katsuzo0828@gmail.com (K.H.); gooddream44@gmail.com (H.W.); dkojima@fukuoka-u.ac.jp (D.K.); watanabemasato@fukuoka-u.ac.jp (M.W.)

**Keywords:** colorectal cancer, tumor microenvironment, T cell receptor repertoire, clonality, diversity, immune checkpoint inhibitor

## Abstract

The rapid increase in colorectal cancer (CRC) cases recently has highlighted the need to use predictive biomarkers to guide therapeutic approaches. Current studies have focused on the tumor-infiltrating lymphocytes present in the tumor microenvironment (TME), in which cytotoxic T cell activation and the amount are associated with CRC patient prognosis. The T cell receptor (TCR) is essential for antigen recognition and T cell identification, playing a central role in cancer immunotherapy. The T cell status reflects TCR diversity or clonality, known as the TCR repertoire. Accordingly, analyzing the TCR repertoire dynamics may help predict the immunological circumstances of the TME in a timely way. In this review, we summarize the TCR repertoire-related knowledge, including its potential use as predictive biomarkers in CRC. The intratumoral TCR repertoire is restricted in CRC patients compared with healthy individuals, as well as in peripheral blood. Patients with deficient mismatch repair display more restriction than those with proficient mismatch repair. Importantly, a higher TCR diversity before treatment and a decrease following treatment may indicate a good response and a better clinical outcome in CRC patients. The future use of TCR repertoire sequencing technology combined with artificial intelligence-based analysis is a potential strategy for CRC therapeutic decision making.

## 1. Introduction

The incidence of colorectal cancer (CRC) is rapidly increasing, making it the third most common type of cancer worldwide [1]. While advanced CRC can lead to bleeding, obstruction, and death, the recent development of multimodal therapies, including immune checkpoint inhibitors (ICIs), has contributed to improved patient outcomes [2,3]. Surprisingly, some patients with cancer cells displaying genetically deficient DNA mismatch repair (dMMR) show a clinical and pathological complete response following ICI treatment [4,5]. Moreover, recent work has indicated that ICIs may also be effective for a subset of proficient MMR (pMMR) patients. However, the number of patients with such “fortune” is limited. Many patients not only show no beneficial effects, but also experience disadvantages, such as ICI-related adverse effects, including endocrine disorders and cardiovascular events [6,7]. Thus, there is a pressing need to identify biomarkers for predicting therapeutic efficacy and selecting suitable candidates for such therapy.

Intrinsically, oncogenesis and most cancer therapies can trigger immune system modulation because the tumor microenvironment (TME) includes numerous immune-related cells, represented as lymphocytes [8,9]. Among them, cytotoxic T cells play a central role in cancer elimination and can be affected by both ICI treatment and conventional chemotherapy and radiotherapy approaches [10]. Accordingly, many investigators have examined T cells in the TME, as a type of tumor-infiltrating lymphocyte (TIL), or peripheral blood, to clarify the relationships between these immune cells and various cancers. For example, CRC is a representative tumor type, with T cell infiltrates serving as a positive prognostic biomarker [11]. Subsequently, recent studies have expanded upon these observations to investigate TIL characteristics and functions using developing sequencing technologies.

Analyzing the T cell receptor (TCR) is a highly debated topic, given that the signaling pathway via TCR engagement with the peptide/major histocompatibility complex (pMHC) is essential for T cell development, activation, maintenance, and exhaustion, and serves as a self-identification mechanism [12]. Importantly, T cell antigen specificity is based on unique TCR sequences, with this diversity or clonality known as the TCR repertoire [13]. Recent advances in high-throughput sequencing technologies have enabled a comprehensive analysis of the TCR repertoire, specifically the amount of different clonotypes and their abundance in T cells, and complementarity determining region 3 (CDR3) sequences [14]. These are also the most studied features of the TCR repertoire, which reflect the activation and response of T cells undergoing clonal expansions.

The correlations between the immune repertoire and the prognosis of various cancers in patients have been frequently studied in the past few decades, providing important therapeutic information [15]. However, the TCR repertoire in CRC has been examined considerably less than in melanoma and blood cancers, which have been investigated the most. In this review article, we summarize the TCR repertoire-related knowledge, including its potential use as predictive biomarkers for ICI therapy, focusing on CRC.

## 2. TCR Basic Structure and Function

The TCR is a membrane-bound heterodimer consisting of two polypeptide chains, αβ or δγ, and is the central protein complex that drives the immune response [16,17]. TCR diversity and antigen specificity are modulated in extracellular domains, including the constant (C), joining (J), and variable (V) domains. Most TCR diversity is generated from the β chain, because it uses an additional diversity (D) segment. Accordingly, TCRβ can provide much higher diversity than TCRα [18,19]. Importantly, hypervariable CDR3 can interact with various peptide antigens and MHC molecules, which is a key component of this specificity [17,20]. Ligand recognition via the TCRαβ chain is transmitted into the cell through the CD3 complex [21], in which TCR/CD3 signaling attenuation contributes to the immaturity and malfunction of T cells [22,23]. During thymic selection, these lymphocytes are differentiated into CD4^+^ or CD8^+^ T cells, depending on the MHC molecule that they have recognized, class II (MHC-II) or class I (MHC-I), respectively [24]. Antigens presented by human leukocyte antigen (HLA)-II molecules activate CD4^+^ T cells, which differentiate into helper T cell subsets, including regulatory T cells (Tregs). Antigens presented by HLA-I molecules stimulate CD8^+^ T cell differentiation into cytotoxic T cells [24]. This TCR signaling pathway is also maintained and modulated by various stimulators, such as cancer and BCG vaccines [25,26].

Neoantigens, also known as tumor-specific antigens (TSAs), are individual and specific antigens generated by the random DNA mutations that occur in cancer cells [27]. Therefore, activated and expanded CD8^+^ T cells induced by neoantigen recognition are cancer specific, with this property being extremely crucial for effective tumor targeting using cancer immunotherapy. Thus, efficient TCR/CD3 structural organization and signaling underlie proper T cell development, function, and specificity.

## 3. Clonality and Diversity of the TCR Repertoire

The TCR provides specificity to T cells, with approximately 10^10^ unique sequences present in humans [13]. When antigen-presenting cells present TSAs to T cells, the T cells become clonally expanded as TSA-specific T cells with unique TCR sequences and decreased diversity [28]. Thus, clonality and diversity are both involved in this process (Figure 1).

The TCR repertoire dynamically changes in the TME, depending on the tumor progression or therapeutic process. Therefore, interpreting the TCR repertoire is important when evaluating the cancer status, such as before or after treatment. Similarly, information in terms of the analyzed lymphocytes should be clarified, including whether their origin was intratumoral or from peripheral blood and whether the DNA or RNA was examined using bulk or single-cell analysis (Figure 2). Most recent studies have employed high-throughput sequencing technologies to comprehensively analyze the TCR repertoire. These methods can also be applied to highly degraded DNA or RNA extracted from formalin-fixed paraffin-embedded samples [29], while previous studies have used traditional quantitative reverse transcription–polymerase chain reaction (qRT-PCR) or flow cytometry techniques. Moreover, the diversity and clonality of the TCR repertoire can be measured by several common indices, such as Shannon’s diversity index, the Simpson index, the inverse Simpson index, and the Gini–Simpson index, with other investigators including additional metrics, such as the highly expanded clone (HEC) ratio [24]. The innovative bioinformatics software available for TCR analysis is also growing [30].

Although various sequencing platforms can be used, a short-read RNA sequence combined with single-cell technology is currently the mainstay among them [30]. Therefore, given the difficulties associated with comparing old and newly published data, we did not distinguish the detailed methodology used for TCR repertoire analysis in the present article.

## 4. TCR Repertoire Differences Between CRC and Healthy Tissues

The TME mainly consists of infiltrating immune cells, notably cytotoxic T cells and Tregs, as well as neovascular cells, stromal cells, and tumor cells [31]. Accordingly, there may be several morphological or functional differences in the TCR expressed on T cells between normal tissues and the TME. Of note, the degree of TIL infiltration is a prognostic factor in CRC, indicating tumor-associated antigen-induced activation of cytotoxic lymphocytes [11]. The capacity to recognize an autologous tumor was limited to approximately 10% of intratumor CD8^+^ T cells [32]. In addition, the TCR repertoire of TILs displays heterogeneity both from an intratumor and inter-patient perspective [33,34]. Thus, the intrinsic capacity of TILs to recognize adjacent tumor tissues is rare and variable.

To our knowledge, limited studies have examined the TCR repertoire differences between CRC tissues and normal colorectal tissue samples from healthy individuals. This is likely because of the difficulty in harvesting normal samples from healthy donors. Previously, Ochsenreither et al. compared the expression patterns of *TRBV* families to quantify the TCR repertoire in tissue samples of normal colonic mucosa and colorectal carcinoma, finding no differences in the restriction between the carcinoma and unaffected colon tissues [35]. Additionally, Song et al. employed the latest single-cell RNA sequencing method to demonstrate that CD4^+^ or CD8^+^ proliferating T cells, CD4^+^ effector memory T cells, CD8^+^ naïve T cells, and Tregs derived from CRC tissues, showed higher TCR clonal expansion than those from matched normal tissues [28]. This increased TCR clonality has also been observed in other malignant diseases, including melanoma, cervical cancer, and lung cancer [36,37,38].

In general, TCR diversity is inversely correlated with age [39]. Consistent with this, early onset CRC (younger than 50 years of age at diagnosis) tumors showed significantly higher TCR diversity in the TILs compared with average onset CRC (over 60 years of age) tumors [40].

## 5. TCR Repertoire Differences Between Intratumoral and Peripheral T Cells

The TCR repertoire can be analyzed as a liquid biopsy from peripheral blood for noninvasive cancer diagnosis [41]. Therefore, several studies have compared the blood TCR repertoire differences between CRC patients and healthy controls. Cao et al. reported that while healthy donors showed evenly distributed *TRBV–J* gene usage and a lower level of TCR repertoire overlaps, CRC patients showed biased *TRBV–J* gene usage and a higher level of TCR repertoire overlaps [42]. In addition, CRC patients showed a significantly higher HEC ratio than healthy donors [42]. This higher TCR repertoire restriction in CRC patient blood samples has also been demonstrated in other studies [35,43,44,45]. Regarding pathology, poorly differentiated CRC cases display a more restricted TCR repertoire, both in peripheral CD4^+^ and CD8^+^ T cells [46]. No correlation was observed between age and TCR diversity in peripheral blood [45].

Subsequently, some studies have investigated TCR repertoire differences between intratumoral and peripheral T cells in CRC patients. Matsutani et al. reported that the TCR repertoire was similar among multiple tissue specimens from different sites of the same CRC tumor, but not in normal tissues or peripheral blood lymphocytes (PBLs) [47]. Additionally, Nakanishi et al. reported that the TCRβ repertoire in TILs showed a significantly higher level of restriction than that in PBLs. Moreover, there was a significantly greater number of CDR3 amino acid sequences in TILs than in PBLs [48]. Another study also identified oligoclonal enrichment of certain TCR sequences in both tumor and blood samples. However, a few TCR sequences with a frequency of >0.1% were commonly detected in tumors [49]. Furthermore, a murine experiment resulted in the CD8^+^ T cell TCR repertoire showing increased clonality in CRC tissues, but not in non-metastatic spleen samples [50].

Collectively, these studies illustrate that CRC patients show a restricted TCR repertoire in peripheral blood, as well as within the tumor. However, the increased clonality in the periphery is less prominent than in the TME. Similarly, the number of tumor-reactive CD8^+^ PD-1^+^ T cells was substantially lower in the periphery than within the tumor, indicating that active immune responses may be localized to the tumor sites [51]. Therefore, although evaluating the TCR repertoire in blood may be available as an alternative method, it simply reflects the trend in immune response circumstances in CRC.

## 6. The TCR Repertoire in CRC Metastatic Sites

CRC often metastasizes to local lymph nodes, then to distant organs, including the liver and lungs [52]. Human cancer-specific cytotoxic T cell lines can be generated from tumor-draining lymph nodes (TDLNs), which play a pivotal role in cancer progression [53]. Therefore, it is also important to elucidate the TCR repertoire within LNs. In CRC patients, metastasis-positive TDLNs showed a significantly lower TCR diversity and more frequent sharing of TCR clonotypes with the primary tumor compared with metastasis-negative TDLNs [54]. Additionally, LN metastasis-positive CRC patients showed a significantly increased TCR diversity, amount of V-J combinations, and immune CDR3 sequences in CRC tissues compared with LN metastasis-negative CRC patients [55]. These results indicate that while cancer-reactive T cells are enriched in metastasis-positive TDLNs, the specific clones are distributed within multiple cancer-bearing tissues, resulting in increased diversity in advanced stage, primary CRC. 

For distant metastasis, Sukegawa et al. revealed that T cells expressing the same TCRβ were clonally expanded in both the primary CRC tumor and metastatic liver site [51]. Moreover, Haraguchi et al. identified shared TCR clonotypes in CD8^+^ T cells between serially resected tissues from primary CRC tumors, liver metastases, and recurrent tumors [56]. Consequently, these results demonstrate that the TCR repertoire is shared between primary CRC and metastatic sites, with memory T cells chronologically preserved in CRC patients.

## 7. The Treg TCR Repertoire

Tregs are master immunoregulatory T cells that are found in the TME [31]. CRC tumors with extensive TIL infiltration also possess highly suppressive Tregs that overexpress *FOXP3* [57]. A murine experiment demonstrated that both the tumoricidal function and clonality of tumor-specific CD8^+^ T cells were prominently modulated by Tregs, underlying the change in *TRBV* gene usage [58]. Although the low proportion of Tregs among the total lymphocyte population has hindered many studies, few have examined the TCR repertoire.

Highly suppressive Tregs present as CD4^+^CD25^+^CD127^low^ T cells in humans [59]. First, Luo et al. evaluated peripheral T cells in metastatic CRC patients. While both CD4^+^ and CD8^+^ T cells showed a polyclonal CDR3 pattern in the healthy control cohort, those in CRC patients showed the restricted CDR3 pattern [44]. Moreover, the CDR3 length distribution indicated more restriction in CD8^+^ T cells compared with CD4^+^ T cells, with prominent gene usage of *TRBV12* and *TRBV16* families within CD4^+^ T cells and *TRBV19* and *TRBV21* families within CD8^+^ T-cells also identified [44]. Zhang et al. developed a single T cell analysis method using RNA sequencing and TCR tracking indices to quantitatively evaluate the dynamic relationships among 20 identified T cell subsets. In CRC tissues, most tumor-infiltrating Tregs with exclusive TCR clonotypes exhibited a high level of clonal expansion, as well as CD8^+^ effector and exhausted T cells, in which Tregs were among the highly expanded populations [60]. Additionally, some intratumoral Tregs shared a TCR repertoire with peripheral blood- and normal tissue-derived Tregs [60].

Similar to that of CD8^+^ T cells, the Treg TCR repertoire in CRC appears to be restricted both in the periphery and the TME. However, further detailed investigations are still needed to uncover their functionality and clinical significance. Perhaps, the in vitro expansion of Tregs using rapamycin may overcome their low population and make them available for future analysis [61].

## 8. TCR Repertoire Differences Between dMMR and pMMR Patients

The most clinically advanced pretreatment biomarkers of ICI responses include CD8^+^ T cell tumor infiltration, intratumoral programed cell death-ligand 1 (PD-L1) expression, the tumor mutation burden (TMB), and the neoantigen burden [18]. Akiyama et al. reported that CRC patients with a higher TMB showed more restricted TCR repertoires than those with a lower TMB [62]. However, in CRC, tumor genetic information, such as the dMMR or pMMR and microsatellite instability-high (MSI-H) or -stable (MSS) status, are particularly important. For example, data have demonstrated that neoadjuvant nivolumab plus ipilimumab treatment can lead to a pathological response in a prominently high proportion of patients with locally advanced dMMR–CRC [5]. MSI-H–CRC cases involve more abundant T cell infiltration than MSS–CRC cases [28]. Accordingly, any difference in the TCR repertoire among these genetic statuses is relevant.

First, Pham et al. clarified that the TILs in MSI-H–CRC cases displayed a significantly lower number of TCRβ clonotypes compared with those in MSS-CRC cases [33]. Similarly, in the NICHE study, where patients with dMMR– or pMMR–CRC received ipilimumab and nivolumab before surgery, dMMR patients showed a significantly higher TCR clonality in CD8^+^ T cells than the pMMR patients at the baseline. However, the TCR clonality was further increased after treatment in both cohorts [6]. Additionally, Borras et al. suggested that MSI-H– and MSS–CRC CD8^+^ T cells showed overlapping phenotypic features, but differed dramatically in terms of their TCR antigen specificities [63]. Furthermore, Inamori et al. investigated the TCR repertoire differences between non-metastatic LNs and primary tumors, finding that significant TCR repertoire overlap was observed in MSI-H/dMMR–CRC patients with a high TMB, while a shared TCR repertoire was limited in MSS/pMMR–CRC patients with a low TMB [64]. Curiously, the same study showed that excessive surgical LN dissection did not have a positive impact on long-term prognosis in MSI-H/dMMR–CRC cohorts [64]. Thus, because regional LNs play an important role in anti-tumor immunity, care should be taken with excessive non-metastatic LN dissection, especially in MSI-H/dMMR–CRC patients.

Consequently, MSI-H/dMMR–CRC patients already show restricted TCR diversity in regard to CD8^+^ T cells equipped with tumor specificity at the baseline, with these tumor-specific T cell clones rapidly expanding following ICI treatment [3]. 

## 9. Can the TCR Repertoire Serve as a Predictor of ICI Efficacy?

Although ICI treatment has demonstrated significant clinical benefits for MSI-H/dMMR–CRC patients, a subset of MSS/pMMR–CRC patients have also shown favorable responses to ICIs [6]. However, selecting candidate individuals for ICI treatment from within these MSS/pMMR–CRC patient groups remains challenging. Because the intrinsic TCR properties determine the fate of CD8^+^ TILs in terms of whether they express PD-1 in the TME [51], the TCR repertoire may therefore be a potential pretreatment biomarker. In melanoma, for example, a higher TIL TCR clonality was associated with the best response to ICIs [65,66], while a higher TCR diversity in the periphery was related to a better ICI response [67]. Moreover, an increased TCR clonality after ICI treatment was observed both in the periphery and within the tumor in melanoma patients [68].

The NICHE study also investigated the TCR clonality differences in terms of CD8^+^ TILs from pMMR–CRC patients who were categorized into ICI responder and non-responder groups. The responder group showed higher CD8^+^PD-1^+^ T cell infiltration and TCR clonality than the non-responder group at the baseline [6]. Interestingly, while the non-responder group displayed a significant increase in TCR clonality following ICI treatment, no significant change was seen in the responder group [6].

Radiotherapy combined with ICI treatment can boost cytotoxic T cell infiltration in the TME, with improved efficacy and increased TCR diversity being reported in a murine experiment [69]. An ongoing phase 2 randomized study that involves MSS–metastatic–CRC patients treated with PD-L1/cytotoxic T lymphocyte antigen-4 (CTLA-4) inhibition, with radiotherapy, has performed TCR repertoire analysis [70]. Although the results have indicated no objective responses beyond the radiotherapeutic effect yet, the CDR3 sequence analysis revealed a significantly increased number of expanded T cells compared with the contracted T cell clones in PBLs. An inverse trend was seen in the TME, where T cell expansion was noted, but was accompanied by a degree of T cell contraction [70]. Notably, the TCR repertoire was found to be clearly overlapping between the TILs and PBLs following the treatment [70].

A phase 2 trial conducted in China also employed TCR repertoire analysis, in which refractory MSS–metastatic–CRC patients were administered fecal microbiota transplantation, plus an anti-PD-L1 antibody and multi-tyrosine kinase inhibitor, as a third-line treatment [71]. Although this study showed improved survival with this treatment approach, no change in the TCR repertoire was observed in the PBLs between the responders and non-responders, nor pre- and post-treatment, while an expanded TCR in the responder group exhibited the characteristics of antigen-driven responses [71].

Taken together, the current studies have not provided sufficient evidence to support the utility of TCR repertoire analysis for predicting the prognosis of pMMR–CRC patients treated with ICIs. However, because these clinical trials all included only a few samples, additional data are required for validation. Additionally, evaluating any TCR repertoire changes is important for any prediction, because it can be dynamically altered depending on the TME. A murine experiment has clarified that treatment with an anti-CTLA-4 antibody initially increased the TCR clonality, with a decrease in diversity, with the effects subsequently gradually subsiding [72]. A subgroup analysis as part of the randomized METIMMOX trial also reported such a transient increase in the monoclonal TCR repertoire following oxaliplatin-based chemotherapy combined with nivolumab [73]. These accumulated studies have shed light on the ICI-induced dynamic changes in the local immune TME, including TCR gene usage, which may help guide future studies and clinical decision-making for the treatment of CRC

## 10. The TCR Repertoire as a Predictor of CRC Prognosis

A higher number of TILs reflects a better prognosis in CRC patients [11]. In addition, LN metastasis-positive patients show significantly increased TCR diversity than LN metastasis-negative patients [55]. Thus, TCR repertoire analysis may help predict CRC patient prognosis.

Sanz-Pamplona et al. reported that a higher TCR abundance and lower clonality in TILs were associated with longer disease-free survival (DFS) in stage II MSS–CRC patients, regardless of whether they have received adjuvant chemotherapy [74]. Campana et al. analyzed the TCR repertoire using CRC datasets in The Cancer Genome Atlas database, finding that patients with tumors that have a high level of TCR clonality lived longer than those with a low level of TCR clonality [57]. The same research group found no significant TCR repertoire differences when comparing right-sided tumors with left-sided tumors and stage I–II tumors with stage III–IV tumors [57]. Furthermore, Mlecnik et al. reported that the levels of *TRBV2* usage in TILs were increased in CRC patients with higher *CX3CL1* gene expression levels, which correlated with longer DFS [75]. Thus, increased *TRBV2* usage in TILs may suggest a better prognosis for CRC patients.

Overall, most studies have reported that high TCR diversity was associated with better disease prognosis, while low diversity correlated with more aggressive phenotypes [76]. In gastric cancer, a higher number of shared TCR sequences was related to a better outcome, but gradually decreased during tumorigenesis [77]. Curiously, TCR repertoire diversity was significantly correlated with the CD8^+^ T cell cytolytic capacity in primary CRC sites [19]. CRC tumors displaying the overexpression of immune response-related genes were associated with increased CD8^+^ T cell infiltration, high TCR diversity, and prolonged survival [78]. Thus, while a higher TCR diversity indicates a functional immune system with a better capacity to orchestrate an anti-tumor response, the loss of diversity may be a consequence of an aggressive tumor, leading to immune system failure [24]. Furthermore, the relative change in TCRβ diversity pre- and post-treatment was associated with longer progression-free survival (PFS) in CRC patients. The diversity of TCRα and TCRβ in blood samples were both decreased after surgery [49], with the TCR clonality in TILs increased after ICI treatment [6]. Thus, analyzing the dynamic TCR repertoire changes at multiple timepoints during CRC treatment may be crucial for evaluating patient prognosis and treatment effectiveness.

## 11. The TCR Repertoire Sequence as a Predictor of Efficacy Using Conventional Multimodal Treatment Methods

In one report, chemotherapy and radiotherapy could not only directly eliminate cancer cells, but also induce subsequent adaptive immune responses against tumors with immunosuppressive mechanisms to escape destruction [79]. Accordingly, analyzing the TCR repertoire may help predict the efficacy of multimodal therapy and CRC patient outcomes.

Tamura et al. clarified that a drastically decreased TCR diversity in CRC tissues during a combination treatment with five cancer peptide vaccines and oxaliplatin-based chemotherapy was associated with longer PFS [49]. Curiously, although the majority of the enriched TCR sequences observed in the tumor prior to treatment were undetectable in the peripheral blood, several enriched clones were detected in the tumor post-treatment [49]. Thus, analyzing the dynamic change in TCR sequencing before and after therapy using peripheral blood may be a useful clinical parameter for monitoring treatment responses.

Previously, Luo et al. analyzed the CDR3 profiles of CD8^+^ T cells from unresectable metastatic CRC patients treated with irinotecan, fluorouracil, and leucovorin (FOLFIRI), with or without bevacizumab. They found that while the patients who showed stable disease or partial remission had a less restricted CDR3 profile after treatment, the patients with progressive disease displayed the opposite result [44]. Recently, Chen et al. reported that TCR diversity and CDR3 clonotypes were decreased in most patients after treatment with FOLFILI with bevacizumab or cetuximab. Moreover, the patients whose TCR diversity dropped remarkably during the treatment showed better clinical responses [45]. Notably, this study also implied that a higher TCR diversity prior to treatment was significantly associated with longer PFS [45]. Additionally, one report demonstrated that a higher level of CD8^+^ TIL clonality was induced by an anti-VEGF antibody compared with an anti-EGFR antibody [80].

The TCR sequence may help predict the efficacy of preoperative therapies, including chemoradiation therapy (CRT) and neoadjuvant chemotherapy (NAC). Akiyoshi et al. quantified the TCR repertoire using pretreatment biopsies from 67 patients with advanced rectal cancer receiving preoperative CRT, revealing that the TCR diversity was significantly higher in good responders compared with non-responders [81]. In addition, larger changes in the TCR repertoire before and after CRT were correlated with better relapse-free survival [81]. The Oslo Randomized Laparoscopic Versus Open Liver Resection for Colorectal Metastases Study employed TCR sequences with resected CRC liver metastasis from 85 patients to compare the repertoires between NAC (regimen was not predefined) exposure and the naïve group. Although no association was observed between TCR clonality and overall survival, the NAC group showed an increased level of TCR clonality [82].

Taken together, a higher TCR diversity at the baseline and a decrease following treatment may indicate a good response to conventional multimodal therapies and better patient outcomes. The potential underlying mechanism is that chemotherapy and/or radiotherapy can modify the TME, possibly by inducing immunogenic cell death and subsequently evoking tumor-specific CD8^+^ T cell expansion, as suggested by a restricted TCR repertoire [73]. These treatments may help CRC tumors potentially become more responsive to ICIs.

## 12. Conclusions

The TCR repertoire can indicate the anti-cancer immunity status of individual patients, which can be transiently altered in concordance with cancer progression or remission. Current high-throughput sequencing technologies with single-cell analysis capabilities have enabled the accurate examination of HLA typing, the clonal expansion of the immune repertoire, and CDR3 motifs in CRC patients. Future studies may sequence individual T cells, obtaining transcriptomic data from T cells expressing TCR, which could be used to analyze cell subtypes and functions. Moreover, if a radiotracer, such as a ^99m^Tc-labeled anti-tumor-specific TCRβ antibody is developed, a sequential, real-time, noninvasive method to visualize T cell dynamics in vivo may be feasible, using single-photon emission computed tomography imaging [83]. Furthermore, by combining accurate HLA typing and artificial intelligence-based structural analysis, TCR–pMHC complexes may be able to be predicted, even before oncogenesis. Identifying the neoantigen and corresponding TCR repertoire, as well as predicting immune responses in the TME, have significant potential in regard to the clinical application of TCR-engineered T cells, including TCR-T and chimeric antigen receptor T cell therapies.

In conclusion, analyzing the TCR repertoire could provide critical information, which may help to guide therapeutic decision making, including in regard to the use of ICIs to treat CRC patients.

## Figures and Tables

**Figure 1 ijms-26-02698-f001:**
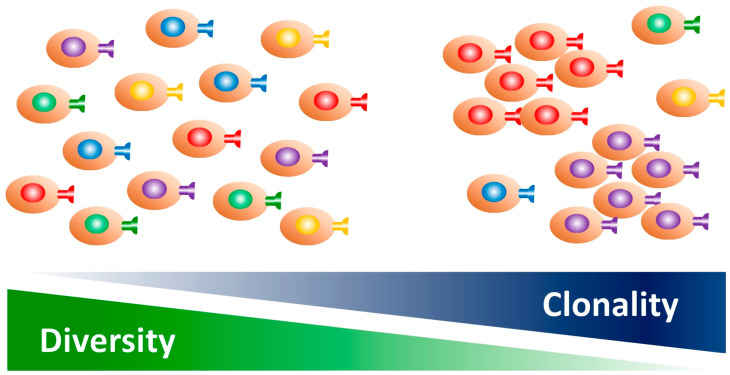
TCR repertoire: diversity vs. clonality are two sides of the same coin. When T cells are presented with tumor-specific antigens (TSAs) by antigen-presenting cells, they are clonally expanded as TSA-specific T cells with a unique TCR sequence, meanwhile their diversity is decreased (i.e., restricted).

**Figure 2 ijms-26-02698-f002:**
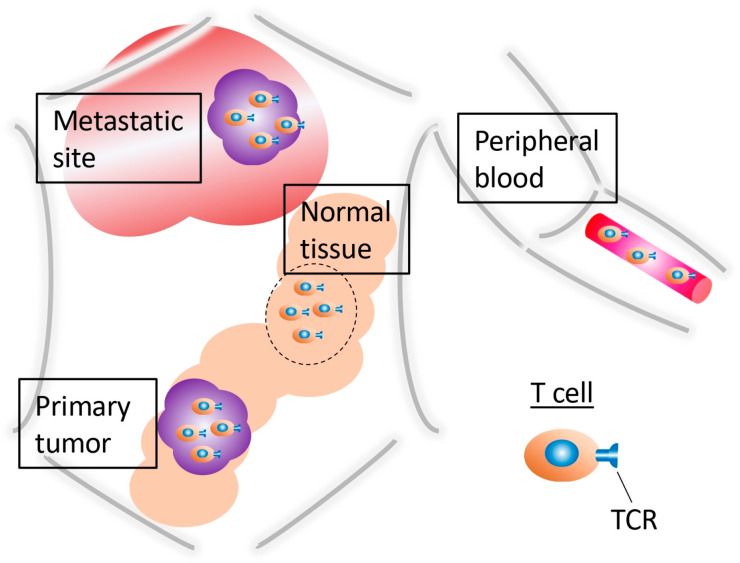
Available samples for TCR repertoire analysis. These are several different types of samples that can be used for harvesting and analysis of the TCR repertoire. However, there are also differences in the TCR repertoires among these samples. The samples are intratumor, metastatic site, peripheral blood, and healthy tissue from colorectal cancer patients.

## Data Availability

No new data were created or analyzed in this study.

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
