# Peer review of "The Current Status of T Cell Receptor (TCR) Repertoire Analysis in Colorectal Cancer"

_ijms, 2025, doi:10.3390/ijms26062698_

Round 1

Reviewer 1 Report

Comments and Suggestions for Authors

This paper is a review of the current state of T cell receptor (TCR) repertoire analysis in colorectal cancer (CRC). Analysis of the TCR repertoire can be useful for predicting treatment effects and for screening patients. The authors have been comprehensively reviewed and present interesting reports.

Important Notes: Unfortunately, Figures 1 and 2 in lines 97 and 103 are stated in the body text, but cannot be verified because the figures and tables are not displayed. The authors should insert the appropriate figures.

It would be useful for readers to understand if the manuscript was edited by paragraph proofreading according to the format of the journal as follows: Introduction, Results, Discussion, and Methods, instead of a series of paragraphs.

In the concluding paragraph, it would be better to not only clarify the significance of CRC biomarkers by TCR repertoire analysis, but also to describe the new technologies that the database will lead to TCR-T and CAR-T therapies in perspective.

Minor: Please correct the typographical errors likeδγ.

Author Response

Comments 1: Unfortunately, Figures 1 and 2 in lines 97 and 103 are stated in the body text, but cannot be verified because the figures and tables are not displayed. The authors should insert the appropriate figures.

Response 1: I’m so sorry but added them now.

Comments 2: It would be useful for readers to understand if the manuscript was edited by paragraph proofreading according to the format of the journal as follows: Introduction, Results, Discussion, and Methods, instead of a series of paragraphs.

Response 2: Thank you for the suggestion. But we considered the current format is better for comprehension because our article is a narrative review, which lack material, methods and results displayed as calculated and summarized data. If systematic review article, we definitely prefer the style you suggested.

Comments 3: In the concluding paragraph, it would be better to not only clarify the significance of CRC biomarkers by TCR repertoire analysis, but also to describe the new technologies that the database will lead to TCR-T and CAR-T therapies in perspective.

Response 3: Pretty nice suggestion! Actually, we wanted to mention about such a evolutional cellular treatment relate to TCR. We added some associated descriptions. Thank you.

Comments 4: Minor: Please correct the typographical errors likeδγ.

Response 4: Sure. Thank you.

Reviewer 2 Report

Comments and Suggestions for Authors

The submitted manuscript, ijms-3514235-peer-review-v1, provides a comprehensive overview of “The Current Status of T Cell Receptor (TCR) Repertoire Analysis in Colorectal Cancer”. The authors have synthesized recent advances in TCR repertoire, particularly highlighting its critical role in CRC therapeutic decision-making. This review has the potential to become a key reference in the field, but requires minor revision before publishing in IJMS.

  • The review provides a focused analysis of TCR repertoire differences between dMMR and pMMR patients. Could the authors discuss whether the correlation between TCR repertoire and tumor mutational burden (TMB).
  • Could the authors explain whether the TCR diversity or its opposite, the clonality can be used as predictors or prognostic biomarkers of the disease?
  • In lines 140-142 of Part 4,why the authors only describe the correlation between TCR diversity and age? How does it correlate with other clinical data?
  • The authors state in lines 183-188, “LN metastasis-positive CRC patients showed a significantly increased TCR diversity compared with LN metastasis-negative CRC patients.” Could the authors discuss whether the increased TCR diversity in LN metastasis-positive CRC patients associated with better prognosis?
  • The authors state in lines 265-267, “while the non-responder group displayed a significant increase in TCR clonality following ICI treatment, no significant change was seen in the responder group.” Could the authors clarify the intended interpretation of this observation?
Comments on the Quality of English Language

The English could be improved to more clearly express the research.

Author Response

Comments 1: The review provides a focused analysis of TCR repertoire differences between dMMR and pMMR patients. Could the authors discuss whether the correlation between TCR repertoire and tumor mutational burden (TMB).

Response 1: So interesting suggestion! We found one study analyzed relationship between TCR repertoire and TMB in CRC. Unfortunately, the article did not clarify MMR status in CRC cohorts but we added some curious information in the current manuscript. Thank you.

Comments 2: Could the authors explain whether the TCR diversity or its opposite, the clonality can be used as predictors or prognostic biomarkers of the disease?

Response 2: Yes. We mentioned about it in Section 10 and 11. Also, we wanted to emphasize the importance of evaluation of the change in TCR repertoire (i.e. harvesting samples repeatedly), rather than analysis at one timepoint.

Comments 3: In lines 140-142 of Part 4,why the authors only describe the correlation between TCR diversity and age? How does it correlate with other clinical data?

Response 3: We could not find other articles that investigated the correlation between TCR repertoires and any other characteristics in CRC patients. We are ready to remove this description from our manuscript if invaluable information.

Comments 4: The authors state in lines 183-188, “LN metastasis-positive CRC patients showed a significantly increased TCR diversity compared with LN metastasis-negative CRC patients.” Could the authors discuss whether the increased TCR diversity in LN metastasis-positive CRC patients associated with better prognosis?

Response 4: I think that is an incisive comment! We have never thought of TCR repertoire within LN metastasis + cohorts in terms of CRC prognosis. Unfortunately, we could not find any studies accessed them, thus, that may be one of the subjects to be uncovered in the future studies. Thank you so much.

Comments 5: The authors state in lines 265-267, “while the non-responder group displayed a significant increase in TCR clonality following ICI treatment, no significant change was seen in the responder group.” Could the authors clarify the intended interpretation of this observation?

Response 5: Thank you for question. This is probably due to the difference in TCR repertoires between responder and non-responder group at baseline: The responder group showed higher TCR clonality than the non-responder group before ICI treatment although not reached at statistical significance. We added these descriptions in the manuscript.

Round 2

Reviewer 1 Report

Comments and Suggestions for Authors

The authors comprehensively reviewed the current state of T cell receptor (TCR) repertoire analysis in colorectal cancer (CRC). Analysis of the TCR repertoire can be useful for predicting treatment effects and for screening patients. 

Figures 1 and 2 with legends are correctly displayed. 

Authors mentioned  prospects of TCR-T therapy based on the significance of CRC biomarkers in lines 403-404.